# Learnability Matters: Active Learning for Video Captioning

**Yiqian Zhang[1], Buyu Liu[2], Jun Bao[2], Qiang Huang[3], Min Zhang[2], Jun Yu[2*]**
[1]Hangzhou Dianzi University
[2]Harbin Institute of Technology (Shenzhen)
[3]National University of Singapore
yiqian.zyq@gmail.com, {buyu.liu, baojun}@hit.edu.cn,
huangq@comp.nus.edu.sg, zhangmin2021@hit.edu.cn, zju.yujun@gmail.com

## Abstract

This work focuses on the active learning in video captioning. In particular, we propose to address the learnability problem in active learning, which has been brought up by collective outliers in video captioning and neglected in the literature. To start with, we conduct a comprehensive study of collective outliers, exploring their hard-to-learn property and concluding that ground truth inconsistency is one of the main causes. Motivated by this, we design a novel active learning algorithm that takes three complementary aspects, namely learnability, diversity, and uncertainty, into account. Ideally, learnability is reflected by ground truth consistency. Under the active learning scenario where ground truths are not available until human involvement, we measure the consistency on estimated ground truths, where predictions from off-the-shelf models are utilized as approximations to ground truths. These predictions are further used to estimate sample frequency and reliability, evincing the diversity and uncertainty respectively. With the help of our novel caption-wise active learning protocol, our algorithm is capable of leveraging knowledge from humans in a more effective yet intellectual manner. Results on publicly available video captioning datasets with diverse video captioning models demonstrate that our algorithm outperforms SOTA active learning methods by a large margin, *e.g.* we achieve about $103\%$ of full performance on CIDEr with $25\%$ of human annotations on MSR-VTT.

## 1 Introduction

Video captioning, which aims to understand videos in the form of describing them in natural language Abdar et al. (2023), becomes a heated task Lin et al. (2022) with the emergence of a large amount of data Li et al. (2023a) as well as transformer-based models Liu et al. (2022). Despite the superior performance, existing methods suffer heavily from the need for time-consuming and labor-intensive human annotations Chan et al. (2020) because of their learning-based nature.

Approaches such as active learning Tharwat and Schenck (2023), semi-supervised learning Yang et al. (2023), and domain adaptation Yu et al. (2023) are proposed to address the above-mentioned data issue. This paper takes the active learning approach where we assume there exists a small amount of labelled data together with a large number of unlabelled ones. Our goal is to select the most informative samples from the unlabelled set and have them annotated by humans such that the video captioning model can achieve the best performance with the minimum human effort. Though uncertainty and diversity have been exploited to investigate the properties of unlabelled data, the

---

*Corresponding author

38th Conference on Neural Information Processing Systems (NeurIPS 2024).

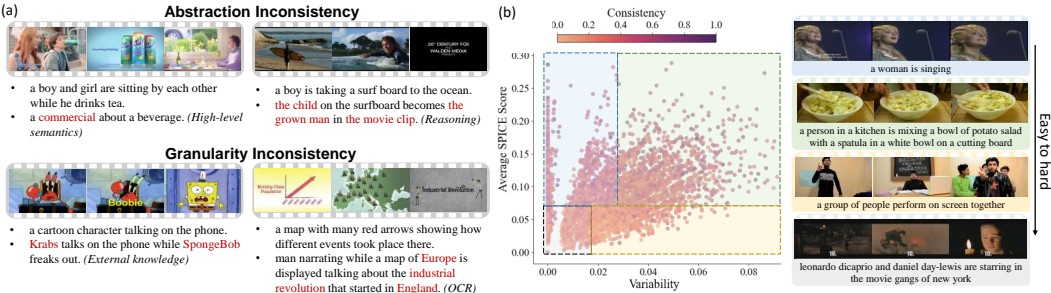

Figure 1: We provide examples of collective outliers as well as their causes in (a). (b) illustrates the Dataset Maps based on the SPICE score for the MSR-VTT training set, with the y-axis representing the average SPICE score over training epochs and the x-axis showing their variability. The consistency score is also measured on ground truth captions. The entire space is divided into four sub-regions based on x,y coordinates, with examples in each region demonstrating different levels of learnability.

potential problem from human annotations is largely neglected in active learning literature as these annotations are not available during the selection process.

In this paper, we first and foremost propose to analyze problems of human annotations in video captioning tasks (see Fig. 1.(b)). With the help of Dataset Maps, we are able to observe the problem, or collective outliers, in human annotations, and conclude its severity by the amount of outliers. Such an amount will further aggravate the learning process due to their hard-to-learn nature, thus worth exploring. We observe that inconsistency in human annotations is one of the main causes of collective outliers, which can be divided into the abstraction and granularity inconsistency (refer to Fig. 1.(a)). The former highlights instances where humans offer different abstractions for videos with complex contents, while the latter reveals that humans may describe the same object at varying semantic levels.

To this end, we incorporate our observations into active learning frameworks. We re-phrase the problem introduced by collective outliers as learnability and aim to address it in our active learning design. Due to the lack of access to human annotations in active learning, directly identifying collective outliers is implausible. Nevertheless, we turn to their main property, or inconsistency, as our breakthrough and propose to estimate the abstraction and granularity inconsistencies in unlabelled data. Concretely, existing image-based large Vision-Language Models (LVLMs) are used to approximate human annotations. Instead of directly using predictions from LVLMs as pseudo ground truths, which would deteriorate the overall performance because of domain gaps, we propose to measure the consistency not only internally among per-frame predictions but externally between predictions from a video captioning model and those per-frame predictions. The internal measurement captures the abstraction consistency and the external one counts on the granularity consistency. Combining both provides us an estimation of sample **learnability**, or how likely this sample belongs to collective outliers. **Diversity** and **uncertainty** are also leveraged in our active learning scheme where we rank samples based on their frequency in the entire dataset and reliability respectively. Our algorithm is able to select reliable, diverse yet learnable samples, and achieves SOTA trade-offs between accuracy and human efforts. Motivated by the causes of collective outliers, we propose a caption-wise active learning protocol such that only a limited number of human-annotated captions are acquired if one video is selected by our active learning algorithm. Our protocol is capable of avoiding inconsistency and providing a more intellectual way to allocate human effort to more diverse videos. In all, our contribution can be summarized as follows:

- To the best of our knowledge, we are the very first in terms of exploring collective outliers on video captioning tasks and providing comprehensive studies on them.
- A novel active learning algorithm that explicitly takes learnability, diversity, and uncertainty into account. Specifically, the learnability is designed to tackle collective outliers, inspired by their abstraction and granularity inconsistency.
- A novel caption-wise active learning protocol that effectively leverages knowledge from humans.
- State-of-the-art performances on video captioning datasets under diverse model backbones.

## 2 Related Work

**Visual Captioning Based on Deep Learning**   Visual captioning Sharma et al. (2023) is a task that automatically generates textual grammatical and semantically appropriate descriptions for a given visual content. While this task is easy for humans, it is extremely difficult for deep learning models. To perform visual captioning, models are required to have multiple capabilities: including but not limited to detection and recognition capabilities to extract objects in visual content, visual semantic understanding capabilities to determine the attributes and relationships of objects, and enriching language knowledge to describe information with grammatically correct sentences. Visual captioning can be classified into image captioning Ming et al. (2022) and video captioning Abdar et al. (2023). Deep-learning-based image captioning starts with the encoder-decoder framework combining convolutional neural networks and recurrent neural networks Kiros et al. (2014); Vinyals et al. (2015). It is further developed by extracting fine object region features Anderson et al. (2018); Karpathy and Fei-Fei (2015), attribute features Yao et al. (2017), semantic relation features Yao et al. (2018); Yang et al. (2022) and introducing attention mechanisms Xu et al. (2015); Pan et al. (2020); Yu et al. (2020). With breakthroughs in vision-and-language pre-training approaches Radford et al. (2021); Zhang et al. (2021a); Alayrac et al. (2022); Li et al. (2023a), applications in image captioning Li et al. (2020); Mokady et al. (2021); Li et al. (2023c) have emerged in recent years, benefit from the power of LVLMs (*e.g.*rich knowledge and strong recognition ability). The recently proposed BLIP2 Li et al. (2023a) has achieved superior performance on major image-text tasks with a simple pre-training method. Unfortunately, LVLMs focusing on video-text-related tasks have not yet come out. Video captioning still faces many challenges. Video captioning is more difficult than image captioning due to the many frames the video contains, which carry massive amounts of information. This raises the necessity of abstracting semantic information from the temporal dimension and describing content from the spatial dimension with different granularity. Video captioning started with the encoder-decoder framework Venugopalan et al. (2015). Subsequent researchers continued to design new encoders Wang et al. (2019); Pan et al. (2016) and decoders Jin et al. (2020); Guo et al. (2016). Some work is also actively trying to introduce scene graphs Hou et al. (2020); Zhang et al. (2020b), text corpus Shi et al. (2023); Zhang et al. (2021b), and pre-training models Li et al. (2023b); Seo et al. (2022); Tang et al. (2021). Among them, Lin *et al.*proposed the end-to-end video captioning method Lin et al. (2022) for the first time and achieved significant performance improvement. Unlike existing video captioning methods, we focus on underlying data issues. Unlike existing video captioning methods, our work focuses on underlying data issues. We are committed to exploiting knowledge to measure video consistency scores for active learning.

**Active Learning**   Due to the high cost of manually annotating data, active learning Tharwat and Schenck (2023) has attracted widespread attention from academia and industry. Active learning aims to use as few, high-quality samples as possible to achieve the best possible performance of the model. Active learning has many mature works in closed tasks such as image classification Parvaneh et al. (2022), object detection Wu et al. (2022), semantic segmentation Xie et al. (2022), and natural language processing Zhang et al. (2022). Among them, Coreset Sener and Savarese (2018) can minimize the distance between an example in the unlabeled pool to its closest labeled example, and can effectively capture the diversity in the dataset. However, active learning for video captioning remains to be explored. Chan *et al*. Chan et al. (2020) tries to migrate conventional active learning methods to video captioning, and has proposed a new method based on ensemble and clustering. Besides, active learning has been observed to fail on open-ended tasks such as visual question answering due to the presence of collective outliers Karamcheti et al. (2021). There is currently no literature that attempts to address the impact of collective outliers. In this work, we focus on the problem of collective outliers in video captioning. We point out possible causes of collective outliers and attempt to mitigate their impact on active learning methods.

## 3 Method

As the very first work that explores data learnability in active learning for video captioning, we start our method with a comprehensive analysis of their causes and various forms in Sec. 3.1. According to our analysis, we then propose a novel active learning scheme that in particular considers the

learnability during the unlabelled data selection process in Sec. 3.2, together with novel designs of uncertainty and diversity. Our overall active learning method can be found in Fig. 2.

## 3.1 Data Learnability in Video Captioning

*Motivation* The order of training samples is shown to be important in curriculum learning Matiisen et al. (2020). Specifically, it found that feeding models with examples of successively increasing difficulty produces better performances than providing full examples immediately. Though posing a more severe problem in active learning as selecting far more difficult samples at early stages is a waste of human efforts, measuring sample difficulty is largely neglected in the literature as neither ground truth nor the definition itself can be easily obtained. To this end, our goal is to first identify difficult examples in video captioning task, and then perform analysis on them so that our observations can be beneficial for our active learning selection scheme.

*Methodology* Collective outliers are a group of data objects that fall extremely far from well-defined norms of a data set or given concepts of expected behavior in data-mining Lai et al. (2021), which are much harder to detect as they are more generic yet harder to identify as individuals. Not surprisingly, one important property of collective outliers is learnability, e.g., they are more likely to be hard-to-learn samples compared to normal data Karamcheti et al. (2021). Concretely, we exploit Dataset Maps to diagnose datasets with training dynamics Swayamdipta et al. (2020); Dagan et al. (2013); Sakaguchi et al. (2020). Compared to conventional Dataset Map that exploits two model-specific metrics (*i.e.*average confidence assigned to the correct answer and the variance of these values) to measure the *learnability* of training examples, we propose to utilize SPICE Anderson et al. (2016) to approximate the confidence score on the correct answer. On the one hand, the confidence score is not a reliable criterion as it can be largely affected by random-length text sequences generated by autoregressive models Zhang et al. (2020a). On the other hand, SPICE measures the F1 score between scene graphs generated by predictions and ground truths. It has the useful property of being defined over tuples that are easy to subdivide into meaningful categories, providing a more semantic-centric evaluation in a human-interpretable manner by abstracting away most of the lexical and syntactic idiosyncrasies of natural language compared to CIDEr Vedantam et al. (2015).

*Observation* We provide an example of Dataset Map of SwinBERT Lin et al. (2022) trained on MSR-VTT Xu et al. (2016) in Fig. 1.(b), where the y-axis and x-axis plots the average SPICE score over training epochs and their variability respectively. Clearly, samples that fall into the upper left area of this figure are easy-to-learn ones whose SPICE scores are consistently high. While the lower left area is occupied with hard-to-learn samples where no observable improvements can be found with longer training time. Unlike conventional close-end tasks where only a small proportion of samples are hard to learn, we observe that a noticeable amount of samples fall into the lower left area.

*Analysis* Taking a closer look at collective outliers, we observe that the video captioning model struggles when learning from samples with inconsistent human annotations. As shown in Fig. 1.(a), these inconsistencies can be divided into two categories, abstraction and granularity. The inconsistency in abstraction summarizes cases where humans tend to provide not only the description of video contents, but also high-level reasoning behind it. For instance, *a boy is walking to the beach to surf* versus *a movie about a man who likes to surf*. On the other hand, inconsistency in granularity consists of samples with human annotations at different graininess,*e.g.*people might coarsely describe the video as *a cartoon character talking on the phone* while a fine-grained description summarizes the same video with *Krabs talks on the phone while SpongeBob freaks out*. We argue that collective outliers are not impossible to learn. Instead, they can be exploited with more consistent descriptions or learned at a later stage of the training process when general principles from easier examples are already learned by models. Our observations and analysis align well with the philosophy of curriculum learning, encouraging our active learning design in the following paragraphs.

## 3.2 Our Active Learning Scheme

Denoting the labeled set as $\mathcal{L} = \{V_m, \mathbf{C}_m\}_{m=1}^{M}$, consisting of $M$ video sequences and their human annotated captions $\mathbf{C}_m$, our video captioning model $f$ is initially trained with $\mathcal{L}$. We further denote another set of $N$ unlabelled videos as $\mathcal{U} = \{V_n\}_{n=1}^{N}$. Mathematically, $\mathcal{U} \cap \mathcal{L} = \emptyset$. Our goal of active learning is to select a subsect $S_t \in \mathcal{U}$ at each selecting step $t \in \{1, \dots, T\}$ such that the overall performance of $f$ can be maximally boosted once annotations on $S_t$ are obtained from humans.

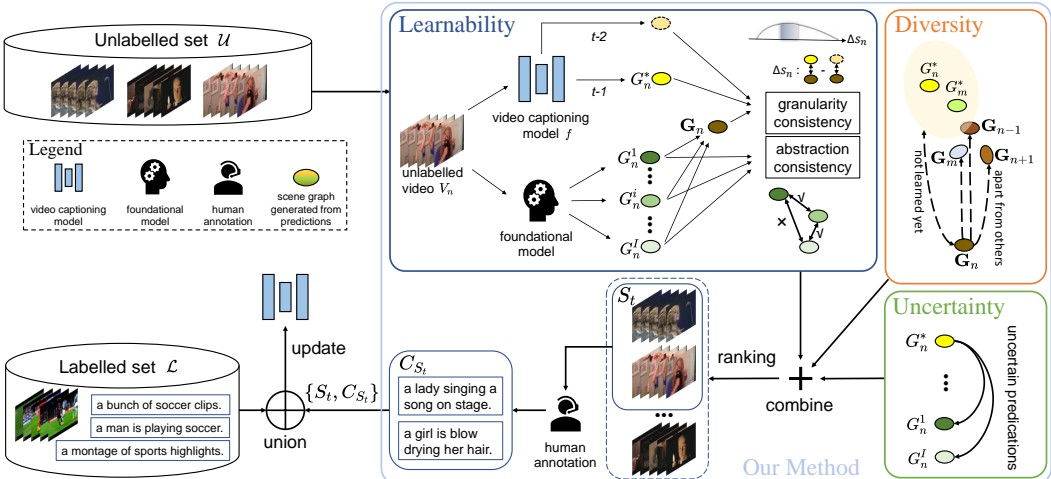

Figure 2: Our method explicitly introduces learnability to reflect the collective outliers in video captioning. Together with uncertainty and diversity, our active learning scheme ranks unlabelled videos and parses them to humans. Our caption-wise protocol further provides an intellectual yet effective way to allocate human efforts, leading to 103% full performances at 25% human annotations.

As stated above, collective outliers can be regarded as a group of hard-to-learn samples in an active learning framework where inconsistencies in human annotations are one of the main causes. Due to the lack of access to these annotations when selecting from $\mathcal{U}$, we propose to mimic them to measure those inconsistencies on the mimicked annotations. In particular, we exploit LVLMs since they provide powerful tools to generate human-like captions at the same granularity. On the one hand, one can approximate the abstraction inconsistency among predictions from LVLMs. On the other hand, by comparing the predictions from $f$ and LVLMs, granularity inconsistency can be further approached. Rather than leveraging foundation models for videos, we utilize the predicted per-frame captions from image-based LVLMs for the following three reasons. Firstly, the image-based LVLMs are more mature and well-exploited compared to the former. Moreover, inconsistency in abstraction occurs more frequently when video samples are complex, e.g., contents vary a lot temporally. Such variety can be better captured by measuring the consistency between frame-wise predictions. Lastly, our design provides a lower bound when measuring the inconsistency in granularity as frame-wise predictions from LVLMs and video captions predicted by $f$ look at videos from different perspectives to the maximum extent, leading to more reliable approximations.

To this end, we uniformly sample $I = 32$ frames from $V_n$ and refer to the $i$-th frame as $V_n^i$. Then we apply BLIP2 Li et al. (2023a) on all sampled frames, leading to a set of $\{b_n^i\}_{i=1}^I$. $b_n^i$ is the predicted caption on $V_n^i$. Meanwhile, we can obtain the video captions on $V_n$ by $f(V_n)$ and further denote the one with the highest confidence as $b_n^*$. Inspired by SPICE Anderson et al. (2016), we convert captions to scene graphs and measure the consistency over graphs. Specifically, we denote $G_n^i = \{O_n^i, A_n^i, R_n^i\}$ as the scene graph generated from $b_n^i$, which consists of objects $O_n^i$, attributes $A_n^i$, and their relations $R_n^i$. Then $\mathbf{G}_n = \{G_n^i\}_{i=1}^I$ is the set of all scene graphs generated from $\{b_n^i\}_{i=1}^I$. Similarly, we can obtain scene graph $G_n^*$ from $b_n^*$. We further denote $\mathbf{O}_n = \bigcup_{i=1}^I O_n^i$, $\mathbf{A}_n = \bigcup_{i=1}^I A_n^i$ and $\mathbf{R}_n = \bigcup_{i=1}^I R_n^i$. In the next few paragraphs, we will introduce our active learning scheme that aims to capture the **learnability**, **diversity** and **uncertainty** in video captioning.

Our scheme is composed of four terms. The first two of them focus on **learnability**, namely abstraction and granularity inconsistencies. The first term $L_n^1$ focuses on the scene graphs generated by per-frame BLIP2, or $\mathbf{G}_n$, and measures their internal prediction consistency. Our $L_n^1$ is defined as:

$$L_n^1 = \frac{\sum_{k \in \mathbf{O}_n} H_k(\mathbf{G}_n)}{|\mathbf{O}_n|} + \frac{\sum_{k \in \mathbf{A}_n} H_k(\mathbf{G}_n)}{|\mathbf{A}_n|}, \tag{1}$$

where $H_k(\mathbf{G}_n)$ counts the number of times that an object, an attribute, or a relationship $k$ appears in $\mathbf{G}_n$. $|\mathbf{O}_n|$ and $|\mathbf{A}_n|$ capture the number of unique objects and attributes in $\mathbf{G}_n$, respectively. Intuitively, Eq. 7 prefers $\{b_n^i\}_i$ when they agree with each other. In other words, the higher the $L_n^1$ is, the more

consistent $b_n^i$ should be w.r.t. predictions from other frames. We purposely neglect relations $R_n^i$ in experiments as the relationship is less reliable in current LVLMs (e.g. BLIP2).

Our second term $L_n^2$ focuses on granularity inconsistency where human annotations are at different graininess. To this end, we approximate the potential granularity inconsistency between $\mathbf{G}_n$ and $G_n^*$ as predictions from image-based LVLMs and that from video captioning model $f$ tend to capture different aspects of $V_n$ to the maximum extent. In particular, granularity inconsistency can be measured by SPICE between two types of predictions, where a lower SPICE value indicates higher inconsistency. Instead of relying solely on SPICE, which tends to select well-learned samples and undermines the active learning scheme, we focus on the time-variant changes in SPICE for each $V_n$ to simulate the expected changes in granularity consistency. Denoting the absolute distance between the SPICE on an unlabelled sample $V_n$ at time step $t-2$ and $t-1$ as $\bigtriangleup s_n$, we have $L_n^2 = g(\bigtriangleup s_n) \log g(\bigtriangleup s_n)$, where $g$ is a min-max normalization function over all $N$ samples. Our $L_n^2$ prefers samples with moderate changes, based on our observation that large and small changes are associated with less informative and hard-to-learn examples, respectively.

The third term $L_n^3$ is designed for **diversity**. Specifically, we prefer unlabelled samples in which contents are beyond the current model $f$ yet of great importance. In practice, we apply the concept of the Term Frequency Inverse Document Frequency (TF-IDF) Robertson (2004) to measure the importance of video content, in which we further incorporate our observation that longer descriptions tend to co-occur more with diverse videos by re-weighting the TF with long captions. Let's denote $\mathbf{G}^*$ as the set that includes predictions of the highest confidence on $\mathcal{L} \cup \mathcal{U}$, or $\mathbf{G}^* = \{G_m^*, G_n^*\}_{n=1,m=1}^{N,M}$. Similarly, the BLIP2 predictions on full dataset is denoted as $\mathbf{G} = \{\mathbf{G}_j\}_{j=1}^{M+N} = \{G_j^i\}_{i=1,j=1}^{I,M+N}$. Mathematically, $L_n^3 = F(\mathbf{O}_n) + F(\mathbf{A}_n)$ where $F(\cdot)$ is defined as:

$$F(x) = \sum_{k \in x} \mathbb{I}[k \notin \mathbf{G}^*] \left( H_k(\mathbf{G}_n) \times \log \frac{N+M}{\sum_j^{N+M} \mathbb{I}[H_k(\mathbf{G}_j) > 0]} \right) \tag{2}$$

where $\mathbb{I}[\cdot]$ is a binary indicator function and equals to 1 iff $\cdot$ is valid. Mathematically, $L_n^3$ focuses only on contents beyond the current model. Meanwhile, it values distinctive samples with important contents. Overall, a higher $L_n^3$ indicates greater diversity in $V_n$.

**Uncertainty** is captured by our last term $L_n^4$. Intuitively, inaccurate predictions provide valuable information in active learning. Therefore, we introduce $L_n^4 = |O_n^* \cap \mathbf{O}_n| + |A_n^* \cap \mathbf{A}_n| + |R_n^* \cap \mathbf{R}_n|$. Specifically, $L_n^4$ counts shared objects, attributes, and relationships between $G_n^*$ and $\mathbf{G}_n$, where $\mathbf{G}_n$ serves as human annotations to measure how well the prediction of highest confidence $G_n^*$ is. A higher $L_n^4$ reflects more certainty in $V_n$.

Finally, the overall active learning score on sample $V_n \in \mathcal{U}$ is defined as $L_n = -\lambda_1 L_n^1 + \lambda_2 L_n^2 - \lambda_3 L_n^3 + L_n^4$, where $\lambda_1, \lambda_2, \lambda_3$ are hyper-parameters. At the $t$-th step of the active learning algorithm, the top unlabelled samples $S_t$ from $\mathcal{U}$ are selected w.r.t. $L_n$, where the lower $L_n$ signifies greater informativeness. These samples are then fed to human annotators to acquire annotations $\mathbf{C}_{S_t}$. Then we update our labelled and unlabelled set with $\mathcal{L} \leftarrow \mathcal{L} \cup \{S_t, \mathbf{C}_{S_t}\}$ and $\mathcal{U} \leftarrow \mathcal{U} \setminus S_t$. Later on, our video captioning model $f$ is re-trained with the updated $\mathcal{L}$. Our overall active learning algorithm iterates until either the maximum time stamp $T$ or human annotation effort is reached.

### 3.3 Caption-wise Selection Protocol

Conventional captioning-related active learning algorithms are video-based where all the human annotations from one video sequence are acquired if this video is selected at step $t$. In practice, there are multiple annotations associated with one video sequence, e.g., we have 20 captions for one video sequence in MSR-VTT dataset Xu et al. (2016). We argue that the current video-based selection protocol is prone to collective outliers, leading to inferior active learning performance. Instead, we propose a caption-wise selection scheme such that not all annotations of one video sequence are acquired if this video has been selected. In practice, we acquire at most 2 [2] captions for each selected video at the $t$-th step. Such a design reflects two of our observations. Firstly, the fewer annotations are acquired from humans, the less likely they are inconsistent with each other, leading to fewer collective outliers. Moreover, including more videos with fewer annotations rather than fewer videos with more annotations boosts the diversity of $S_t$. The superiority of our protocol is shown in Sec. 4.

---

[2]This number is chosen by experiment.

# 4 Experiment

We validate our ideas with various backbones on two publicly available datasets MSVD Chen and Dolan (2011a) and MSR-VTT Xu et al. (2016), and demonstrate SOTA performances on both.

## 4.1 Dataset and Experimental Setup

**Datasets** We conduct our experiments on two datasets, MSVD Chen and Dolan (2011a) and MSR-VTT Xu et al. (2016). Specifically, MSVD consists of 1970 open-domain videos collected from a commercial video search engine, each of which is associated with about 41 human-annotated captions. Similarly, there are 10K open-domain videos in MSR-VTT and each of them has 20 human-labelled annotations. For each dataset, we follow their standard splits and report our active learning performance on their test sets. To mimic the learning process, we initialize $\mathcal{L}$ with $5\%$ of data randomly selected from the training set, including both videos and their annotations. Consequently, $\mathcal{U}$ is composed of videos from the remaining $95\%$ training set. At each selecting step, the annotation budget $\|\mathbf{C}_{S_t}\|$ is set to $5\%$ captions of the full training set. More details can be found in the appendix.

**Baselines** Active learning for video captioning is under-explored in literature. We follow the existing work Chan et al. (2020) and compare our algorithms with the baselines described below:

- **Random Sampling** serves as a very competitive baseline in open-ended tasks. Basically, it performs random sampling in $\mathcal{U}$ to obtain $S_t$.
- **Maximum Entropy** Chan et al. (2020) is a conventional uncertainty-based active learning algorithm where samples with the highest entropy will be selected. In video captioning tasks, the entropy of sample $V_n$ is approximated by the averaged entropy over multiple predictions in $f(V_n)$. For each prediction, its entropy is computed over word output distributions at each new word.
- **Minimum Likelihood** Chan et al. (2020) is another uncertainty-based active learning algorithm. Typically, samples with the lowest log-likelihood will be selected. In practice, the averaged log-likelihood over multiple predictions in $f(V_n)$ is used to rank each unlabelled video $V_n$.
- **Core-Set Selection** Sener and Savarese (2018) is a classical diversity-based active learning algorithm. Specifically, it works on the representation space and aims to select samples that are spread out in the feature space. In practice, features from Liu et al. (2022) are used to select samples that minimize the distance between the unlabelled pool to its closest labeled sample.
- **Clustered-Divergence** Chan et al. (2020) ensembles multiple models and computes the KL-divergence between the conditional distributions of the ensemble members to measure sample uncertainty. Diversity is also implicitly considered as it already ensembles a clustering-based active learning regularization method.

**Implementation Details** In our experiment, we employ SwinBERT Lin et al. (2022) and CoCap Shen et al. (2023) as our $f$ as they provide good trade-offs between accuracy and efficiency. In contrast, other SOTA video captioning methods or video-based foundation models, such as COSA Chen et al. (2023) and mPLUG-2 Xu et al. (2023), require extensive pre-training or finetuning when adapting to downstream tasks to achieve SOTA video captioning performances. To further speed up the training process, each video is uniformly sampled with 32 frames and these frames are parsed to $f$. The batch size is set to 4. We refer to the performance of $f$ that trained with the entire training set on the test set as *full performance*. And we are able to achieve comparable performance compared to the official release. As for BLIP2 Li et al. (2023a), we choose an open-sourced version [3]. All experiments are conducted on 4 RTX 3090 GPUs and 4 RTX 4090 GPUs with Pytorch Paszke et al. (2019), Huggingface transformers Wolf et al. (2020). $T$ is set to 4 . For other hyper-parameters, we keep the same configuration as in Das et al. (2022). More details can be found in the appendix.

*Evaluation Metrics* We provide detailed comparisons using a diverse set of performance metrics, including BLEU4 Papineni et al. (2002), METEOR Banerjee and Lavie (2005), ROUGE-L Lin and Och (2004), CIDEr Vedantam et al. (2015), and SPICE Anderson et al. (2016).

## 4.2 Main Active Learning Performances

We report the active learning performances of all methods with either SwinBERT or CoCap on MSVD in Fig. 3. To ensure more reliable results, all methods were conducted three times. This figure reports both the average performance and the variance.

---

[3]Checkout the pre-trained model at `https://huggingface.co/Salesforce/blip2-flan-t5-xl`

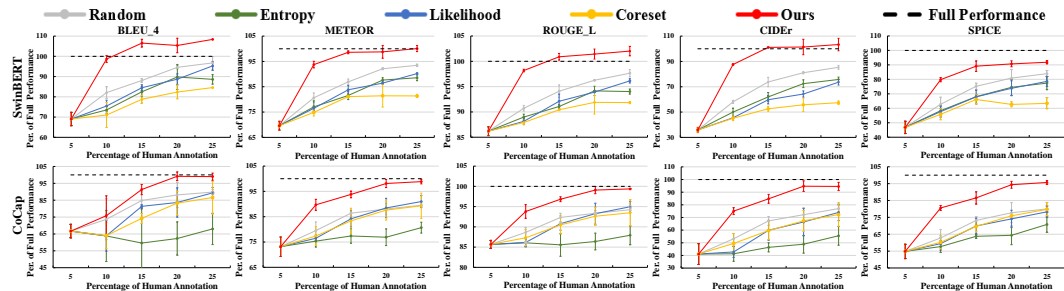

Figure 3: Active learning performances on the MSVD. Ours significantly outperforms other methods.

There are several interesting observations. First and foremost, it is noteworthy that our method consistently outperforms all other baselines across all five evaluation metrics with two backbones, highlighting our superiority. Predictably, random sampling usually ranks as the second-best option, aligning with findings from other open-ended tasks Karamcheti et al. (2021). More importantly, we observe that our method surpasses the full performance of $f$ trained with $100\%$ annotations with SwinBERT out of 4 evaluation metrics, as indicated by the dotted horizontal line, with less than $25\%$ of annotations. For example, we achieve full performance with about $10\%$ and $15\%$ of annotations when measured by BLEU4 and ROUGE-L, respectively. Regarding CIDEr, our method attains $103.65\%$ of the full performance with $25\%$ of annotations. Even with CoCap, we can almost always achieve more than $95\%$ of full performance with $25\%$ annotations. This observation supports our concerns about the negative impacts of collective outliers during the training process. Clearly, our method effectively reduces the impact of collective outliers, leading to significant performance improvements over existing methods and even surpassing full performance. Results on MSR-VTT are reported in the appendix. In summary, our method significantly outperforms the SOTA methods and achieves an averaged $103\%$ full performance with $25\%$ annotations on all metrics.

**Cross-dataset Performance** To simulate the scenario where people tend to exploit a large unannotated dataset to benefit a small annotated dataset, we use the small annotated dataset MSVD (1,200 videos with 40 captions per video) and treat MSR-VTT as the large unannotated dataset, which includes 6,513 videos paired with 20 captions each. The results using the official SwinBERT implementation are provided in Tab. 1.

Table 1: Starting from fully-annotated MSVD training samples, exploiting data from MSR-VTT with our AL algorithm can further boost the performances on the MSVD test set.

| Method | Data Per. | BLEU_4 | METEOR | ROUGE_L | CIDEr | SPICE |
|---|---|---|---|---|---|---|
| Starting Point | 0 | 55.71 | 39.70 | 75.73 | 109.39 | 6.97 |
| Random | 20 | 62.15 | 42.58 | 78.66 | 123.26 | 7.63 |
| Ours | 20 | 63.70 | 43.69 | 79.86 | 127.41 | 7.80 |
| Ours | 5 | 63.68 | 43.34 | 79.73 | 126.32 | 7.57 |
| Ours | +5 | 63.10 | 43.64 | 79.98 | 130.96 | 7.89 |
| Ours | +5 | 63.51 | 43.80 | 79.81 | 129.63 | 7.93 |
| Ours | +5 | 64.88 | 44.25 | 80.43 | 129.08 | 7.77 |

We report the overall performance on the MSVD test set. "Data Per." is the percentage of human annotations on MSR-VTT. We also report the performance of directly selecting $20\%$ of MSR-VTT (Row 4) and iteratively adding $5\%$ of MSR-VTT four times (Rows 5-8). As shown in the table, our AL algorithm significantly improves the overall performance of MSVD and is a more effective choice compared to random selection. Furthermore, directly selecting $20\%$ of data is slightly less effective than iterative selection, demonstrating the benefits of curriculum learning. Notably, the overall performance peaks at two iterations, or $10\%$ of human annotations on MSR-VTT, according to CIDEr and SPICE. Beyond this point, the performance saturates and slightly declines. This is expected, as $20\%$ of MSR-VTT includes 26K captions and at least 1.3K videos, which is comparable to the original training set of MSVD. Adding more data from a different dataset can degrade performance, as the training may deviate from the original dataset.

### 4.3 Ablation Study on Learnability, Uncertainty, and Diversity

To demonstrate the effectiveness of individual components of our design, we conduct a thorough ablation study by incrementally integrating various components into our active learning scheme. The steps are as follows: 1) $L_n^4$ (uncertainty); 2) $+L_n^3$ (diversity); 3) $+L_n^2$ (learnability); 4) $+L_n^1$ (learnability); 5) + Caption-wise Protocol (CP); Specifically, we report the Area-Under-Curve(AUC) score over CIDEr and SPICE curve in Tab. 2. Evidently, incorporating any of these components will enhance overall active learning performance, demonstrating their effectiveness. For example, our designed terms outperform Random Sampling significantly, indicat-

Table 2: Ablation study on MSVD.

|          | $AUC_{CIDEr}$ | $AUC_{SPICE}$ |
|----------|---------------|---------------|
| Random   | .574          | .560          |
| $L_n^4$  | .583          | .578          |
| $+L_n^3$ | .589          | .585          |
| $+L_n^2$ | .594          | .592          |
| $+L_n^1$ | .603          | .600          |
| +CP (Ours) | .738        | .678          |

ing the effectiveness of our active learning scheme without CP. Notably, we observe a significant performance boost after integrating the CP, which is reasonable as it directly reduces the impact of collective outliers and improves overall diversity.

To further demonstrate that learnability, uncertainty, and diversity each reflect distinct aspects of active learning, we report the percentage of overlapped selections in $S_1$ under active learning schemes focused on individual aspects, as shown in Fig. 4. As expected, our design targeting learnability, uncertainty, and diversity addresses various unlabelled samples, leading to limited overlap in their output $S_1$. Together with Tab. 2, we conclude that all aspects are complimentary and mutually informative.

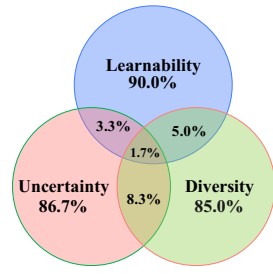

### 4.4 More Analysis

**Are we celebrating from more annotations per video?** As summarized in Sec. 3.1, ground truth inconsistency is one of the main causes of collective outliers. We further showcase in Fig. 3 that fewer human annotations bring in better performance. Then a natural question to ask is whether we are celebrating from more annotations per video.

Figure 4: The percentage of overlapped selections in $S_1$.

To answer this question, we conduct a caption-wise random sampling experiment with SwinBERT on MSR-VTT. Specifically, SwinBERT is trained with all training videos on MSR-VTT where each video is associated with $k \in \{1, 3, 5, 7, 9\}$ captions that are randomly sampled from a full set of 20 human annotations. We visualize the CIDEr and SPICE score on the test set of MSR-VTT in Fig. 5. As shown in the figure, full performance is achieved with approximately 35% of the annotations (equivalent to $k=7$). Overall performance improves with up to nine annotations. We argue that the performance degradation with the full training set is not due to overfitting, as techniques such as regularization, data augmentation, and early stop-

ping have been applied to mitigate it. Instead, we hypothesize that it is related to collective outliers, where inconsistency increases with the number of annotations. Additionally, we observe that caption-wise random sampling performs worse than ours with the same amount of human annotations, highlighting the effectiveness of the $L_n$ design.

**Can we truly reduce the impact of collective outliers?** Though we leverage the knowledge from LVLMs to approximate human annotations and thus identify collective outliers, it remains unknown how well such approximation or identification is. To address this, we first divide unlabelled

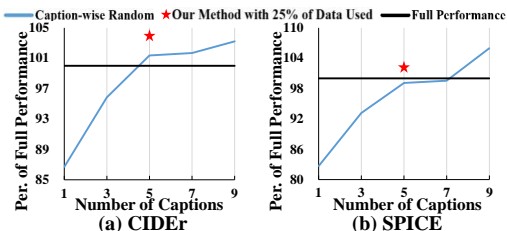

Figure 5: The performance of SwinBERT when trained on samples equipped with different numbers $k$ of annotation captions.

samples into different groups w.r.t. their learnabilities, and then we obtain the distribution of $S_t$ according to these groups. Ideally, our method should select fewer samples that belong to collective outliers. Specifically, we divide unlabelled samples into four discretized groups according to their $x, y$ coordinates in Dataset Maps in Fig. 1. (b). Easy samples are those whose $y > 0.07$ and $x < 0.03$.

Moderate ones fall into the region with $y > 0.07$ and $x \geq 0.03$. For these samples whose $y \leq 0.07$ and $x > 0.17$, we call them hard samples. The remaining samples are then collective outliers.

We report the data distribution of $S_1$ of all methods in Fig. 6. Again, results are obtained on MSR-VTT with SwinBERT as $f$. First and foremost, we observe that our method is able to select the least collective outliers compared to other baselines, which fulfills our goal as expected. Another interesting observation is that our method prefers to select more easy samples at $S_1$. It is an effective strategy to select easy and moderate samples for learning in the early stage of model training, which greatly improves the efficiency and effect of training. By avoiding collective outliers and more reliable samples, it's no wonder ours achieves SOTA performance. We refer the readers to the appendix for more analysis.

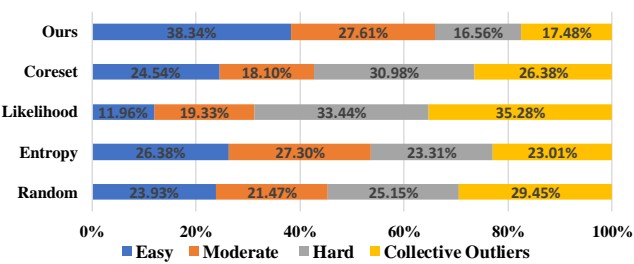

Figure 6: The distribution of samples in $S_1$ on MSR-VTT.

**Can we exploit the knowledge from LVLMs more?** Another way to utilize the predictions from LVLMs is to directly apply them as pseudo ground truths to update $f$ in a semi-supervised learning setup. Specifically, this simple approach achieves 0.31 and 0.07 for CIDEr and SPICE on MSR-VTT, which are worse than the starting points. Adding additional control, *e.g.* including only videos when predictions from LVLMs are highly consistent, achieves 0.5185 and 0.0712 for CIDEr and SPICE, which is worse than our algorithm. We refer the readers to the appendix for more details.

**Are inconsistencies in ground truths our illusions?** To validate our hypothesis that inconsistency in human annotations genuinely exists and is not merely due to subjective judgments, we utilize ChatGPT Radford et al. (2019) as an objective tool. We observe that ChatGPT believes that 49% of all captions are inconsistent on average. Such inconsistency rate increases when moving from easy to collective outlier sub-regions. More details can be found in the appendix.

**Does the reduction of human annotation costs justify the extensive computational cost?** Our additional computational costs arise from the active learning algorithm, primarily due to applying BLIP2 to the unlabelled dataset and generating scene graphs from its predictions. This process occurs only once on the unlabelled set. On our hardware, consisting of 4 RTX 4090 graphics cards with a power capacity of 2000 kWh, it takes no more than 9 minutes to run BLIP2 and 5 minutes for scene graph generation on the MSVD training dataset. In contrast, annotating the full dataset in 2010 required hundreds of annotators, around 2 months, and less than 5000 USD in total Chen and Dolan (2011b). Therefore, we argue that active learning is more efficient in terms of both time and cost.

**Limitations** There are several limitations that we believe are worth further exploration. Firstly, our paper only briefly touches on the relationship between curriculum learning and learnability. Beyond provoking the design of the learnability term, we believe curriculum learning can enhance the interpretability of learnability terms and even active learning. Secondly, we found that current evaluation metrics, such as CIDEr, do not always align with human evaluations. More human analysis is needed for video captioning tasks. Thirdly, we made some preliminary attempts to combine knowledge from LVLMs in a semi-supervised learning manner. Although we did not see a significant improvement, we believe further efforts are warranted. Lastly, our experiments with ChatGPT-4 can be improved with more refined designs. We will include these limitations in our final version.

## 5   Conclusion

In this work, we propose a novel active learning algorithm for video captioning tasks, which effectively leverages learnability, diversity, and uncertainty. To the best of our knowledge, our algorithm is the very first one that targets collective outliers in video captioning and further proposes to reduce their impacts by measuring sample learnability, as well as introducing a caption-wise protocol. Results on two datasets demonstrate the superiority of our algorithm over SOTA methods, *e.g.* we can achieve 103% of full performance with 25% of human annotations on MSR-VTT.

# 6 Acknowledgement

This work was supported in part by the National Natural Science Foundation of China (No. 62125201, 62020106007).

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

# A  Appendix

In our appendix, we first share our quantitative results on the MSR-VTT Xu et al. (2016) dataset in Sec. A.1, followed by more details about our implementation in Sec. A.2. We further include our comprehensive analysis of the property of Dataset Maps, knowledge from LVLMs, and results with ChatGPT in Sec. A.3, Sec. A.4, and Sec. A.5, respectively. Our code and model will be made available.

## A.1  Results on MSR-VTT

We report the overall performances on MSVD in Fig 7 where methods are visualized with different colors. As can be found in this figure, results on MSR-VTT share similar trends with those on MSVD. First and foremost, our proposed method consistently surpasses all baselines under all evaluation metrics. Among them, random sampling remains the safest choice as it almost always is the second-best. Both observations verify our motivation for tackling collective outliers in open-ended tasks. Remarkably, our proposed active learning method quickly achieves full performance using only a few annotations. For example, our method achieves more than $90\%$ of the full performance under four evaluation metrics including BLEU4, METEOR, ROUGE, and CIDEr, with only $10\%$ annotations. Moreover, it achieves $107\%$ of the full performance under CIDEr, when only $25\%$ of the full annotations are exploited. We would like to note that our hyper-parameters in $L_n$ are learned on MSR-VTT only and directly applied to MSVD, which further showcases the robustness and generalizability of our proposed method.

## A.2  Implementation Details

### A.2.1  More Details of Baselines

As described in our main paper, we include two uncertainty-based active learning baselines as suggested in Chan et al. (2020), or Maximum Entropy and Minimum Likelihood, for video captioning tasks. More details of them are provided in the following.

Given a video input $V_n$, the output of the video captioning model $f$, or $f(V_n)$, comprises a set of $K$ generated captions $\mathbf{Y}_n$ with length $W$. We denote the $k$-th caption in $\mathbf{Y}_n$ as $y_{n,k}$ and the $w$-th word in $y_{n,k}$ as $y_{n,k}^w$. Therefore, $\mathbf{Y}_n = \{y_{n,k}^w\}_{k=1,w=1}^{K,W}$. During inference, these $W$ words are obtained in a sequential manner, where the prediction of the $w$-th word relies on its $w-1$ predecessors. Mathematically, we can obtain the conditional probability of the $w$-th word in the $k$-th caption $P(y_{n,k}^w|y_{n,k}^{w-1}, \cdots, y_{n,k}^1, V_n; f)$, representing the likelihood of generating the $w$-th word given the previous words as well as the input video. Before introducing the two baselines, let's further denote the joint distribution of the $W$-th word in the $k$-th caption by $P(y_{n,k}^W) = \prod_{w=1}^{W} P(y_{n,k}^w|y_{n,k}^{w-1}, \cdots, y_{ni}^1, V_n; f)$.

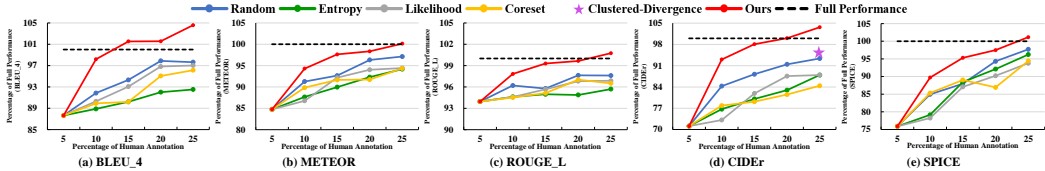

(a) BLEU_4     (b) METEOR     (c) ROUGE_L     (d) CIDEr     (e) SPICE

Figure 7: Active learning performances on MSR-VTT with Swinbert. The y-axis shows the percentage of the full performance under various evaluation metrics, which is obtained by training $f$ with $100\%$ training data. Our proposed algorithm outperforms all other methods by a large margin. Similar to what we observed on MSVD, our algorithm surpasses the full performance with only $25\%$ of annotations. Numbers of Clustered-Divergence Chan et al. (2020) come from the original paper [6]. In particular, it is able to achieve 95% of the full performance with 25% of the annotations with Transformer mode Chan et al. (2020).

**Maximum Entropy** To approximate the expected entropy of $V_n$ in video captioning tasks, the averaged entropy over $\mathbf{Y}_n$ is utilized. For each prediction $y_{n,k} \in \mathbf{Y}_n$, its entropy is computed over word output distributions at each new word. Mathematically, the acquisition function of Maximum Entropy Chan et al. (2020) is defined as:

$$L_n = \frac{1}{|\mathbf{Y_n}|} \sum_{k=1}^{K} \sum_{w=1}^{W} -P(y_{n,k}^w) \ln P(y_{n,k}^w), \tag{3}$$

At each time step $t$, Maximum Entropy selects the unlabelled videos with high $L_n$, reflecting the intuition that samples with high uncertainty are more likely to be informative.

**Minimum Likelihood** In contrast, Minimum Likelihood directly focuses on the averaged probabilities rather than entropy. Specifically, we follow the literature Chan et al. (2020) and define the acquisition function of Minimum Likelihood as:

$$L_n = \frac{1}{|\mathbf{Y_n}|} \sum_{k=1}^{K} \sum_{w=1}^{W} \ln P(y_{n,k}^w | y_{n,k}^{w-1}, \cdots, y_{n,k}^1, V_n; f), \tag{4}$$

Similarly, Minimum Likelihood favors unlabelled samples with low $L_n$, which is well-aligned with the assumption that uncertain samples are more valuable.

### A.2.2 More Details of Our Method

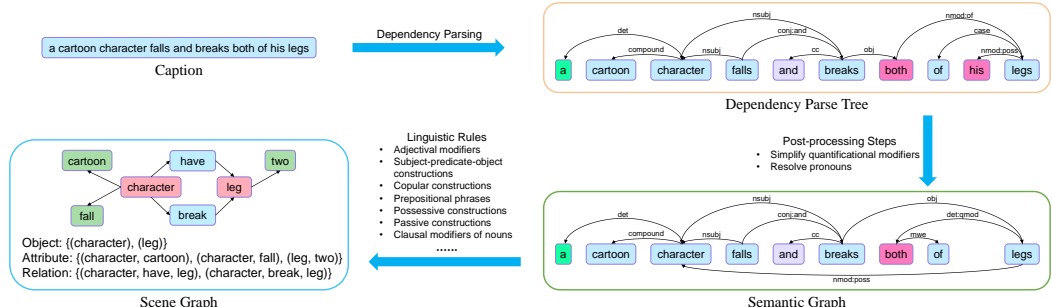

Figure 8: Pipeline of the scene graph parser.

**Scene Graph Parser** Scene graphs are of great importance to our method as our learnability, uncertainty, and diversity are all measured upon them. Therefore, we describe the scene graph parser in length here. Specifically, We reused the pipeline of SPICE Anderson et al. (2016) to generate scene graphs from caption predictions, in which a Stanford Scene Graph Parser Schuster et al. (2015) is adopted. In particular, given a caption $b_n^i$ or $b_n^*$, a Probabilistic Context-Free Grammar (PCFG) dependency parser Klein and Manning (2003) aims to firstly translate it into dependency parse trees. Then a semantic graph can be obtained by simplifying quantificational modifiers and resolving pronouns. After parsing the semantic graph through some simple linguistic rules, a scene graph $G_n^i$ or $G_n^*$, which consists of objects, their attributes, and relationships, can be obtained. We provide an example of one caption and its scene graph in Fig.8.

**Selection Protocols** As formulated in the main paper, our acquisition function for the unlabelled sample $V_n$ is:

$$L_n = -\lambda_1 L_n^1 + \lambda_2 L_n^2 - \lambda_3 L_n^3 + L_n^4, \tag{5}$$

where $\lambda_1, \lambda_2, \lambda_3$ are hyper-parameters. Let's further denote the annotation budget, or the amount of annotations to be obtained, at the $t$-th step of the active learning algorithm as $\mathbf{A}_t$. In our case, $\mathbf{A}_t = \|\mathbf{C}_{S_t}\|$.

**Video-wise Selection** Under conventional video-wise selection protocol, once $V_n \in \mathcal{U}$ is selected by active learning algorithms, all annotations in $V_n$ will be acquired. Assuming that each $V_n$ is associated with $D$ captions, the video-wise selection protocol then selects the top $\|S_t\| = \frac{\mathbf{A}_t}{D}$ videos from $\mathcal{U}$ and get all their captions. In practice, $D$ equals to 20 and 40 in MSR-VTT Xu et al. (2016)

and MSVD Chen and Dolan (2011a). As a reference, we denote the annotations for the $m$-th labelled video $V_m$ as $\mathbf{C}_m$. Denoting the $d$-th caption as $C_m^d$, our $\mathbf{C}_m$ can be represented as $\mathbf{C}_m = \{C_m^d\}_{d=1}^D$.

**Caption-wise Selection** Rather than assuming that all $D$ captions will be acquired once $V_n$ is chosen by active learning algorithms, our caption-wise selection scheme allows more flexibility w.r.t. Eq. 5. Specifically, at each selection step, the video that has been least annotated with lower Eq. 5 will be chosen to get captions. Mathematically, we make the following revision:

$$\hat{L}_n = L_n + \frac{|\mathbf{C}_n|}{q}, \tag{6}$$

where $\mathbf{C}_n$ denotes the number of annotations that already obtained on $V_n$. And $q$ is a dataset-specific hyper-parameter served as a re-ranking factor. Eq. 6 reflects our intuition that videos that have not been annotated are more informative due to both diversity and inconsistent annotations from collective outliers. Consequently, we will rank all $V_n$ according to $\hat{L}_n$ as long as $|\mathbf{C}_n| \neq D$ with our caption-wise selection scheme.

After ranking all $V_n$ that has not been fully annotated, we then start to acquire human annotations w.r.t. $\mathbf{A}_t$. Instead of equally allocating human efforts to top-ranking videos, which neglects the learnability property, we further introduce an intellectual design where more budgets are provided to topper-ranking videos. Compared to their peers, these topper-ranking ones are less likely to be collective outliers. Therefore it is safer and more effective to spend more effort on them. Specifically, we propose a stage-wise acquisition scheme by dividing the ranked videos into $R$ consecutive and exclusive regions according to their $L_n$. If the ranked $V_n$ belongs to the $r$-th region, where $r \in \{1, \ldots, R\}$, then we allocate $A_{t,r}$ annotations to it. Let's denote annotations $V_n$ get at time step $t$ as $\mathbf{C}_n^t$. In this case, $A_{t,r} = |\mathbf{C}_n^t|$. As a reference, we have $\mathbf{A}_t = \sum_n |\mathbf{C}_n^t|$ and $\mathbf{C}_n = \bigcup_t \mathbf{C}_n^t$. Our full caption-wise algorithm is summarized in Alg. 1.

**Hyper-parameter Setting** We set the number of sampled frames $I$, the total active learning step $T$, and the per-step annotation budget $\mathbf{A}_t$ to 32, 4, and 5% of full annotations in each dataset. The hyper-parameters $\lambda_1, \lambda_2, \lambda_3$ in Eq. 5 are 3, 1, and 2, respectively. And they are chosen based on experiments on the validation set via grid search. Our re-ranking factor $q$ in Eq. 6 is set to 10 on MSR-VTT. Meanwhile, $R$ is set to 3, dividing the ranked videos into 3 regions according to $\hat{L}_n$. Specifically, both the first and last regions consist of 2000 videos. $A_{t,r}$ equals to 2, 1, and 0 with $r = \{1, 2, 3\}$. On the MSVD dataset, $q$ and $R$ are set to 20 and 5. These 5 regions in MSVD consist of 250, 250, 250, 188, and 262 unlabelled videos, respectively. With increasing $r$, we have $A_{t,r}$ set to 4, 3, 2, 1, 0.

We would like to note that hyper-parameters related to the Caption-wise Selection Protocol are not optimal due to the lack of tuning. We might expect performance boosts with a more careful search.

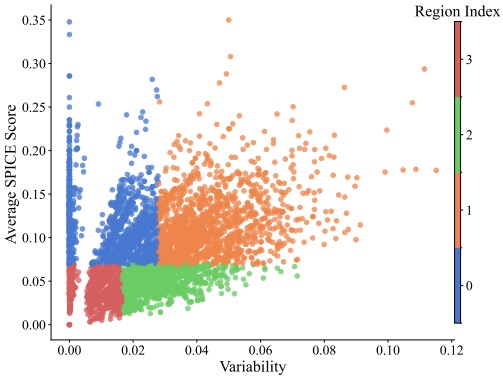

Figure 9: We divide the data in Dataset Maps on MSR-VTT into four regions according to their learnability. Specifically, we have EASY, MODERATE, HARD, and COLLECTIVE OUTLIER colored in blue, orange, green, and red, respectively.

## A.3 Analysis on Dataset Maps

To the best of our knowledge, we are the very first to address the collective outliers in video captioning tasks. As described in our paper, we adapt Dataset Maps Swayamdipta et al. (2020) to perform analysis on collective outliers. We provide Dataset Maps on the training set of MSR-VTT in Fig. 9. In particular, $y$ and $x$ axis are the averaged SPICE score over all training epochs and their variance. Intuitively, the average SPICE score reflects sample difficulty, while the variance represents its

**Algorithm 1:** Our Caption-wise Algorithm

---

**Input:** The labeled set $\mathcal{L} = \{V_m, \mathbf{C}_m\}_{m=1}^{M}$, the unlabelled set $\mathcal{U} = \{V_n\}_{n=1}^{N}$, a video captioning model $f$, a foundational model $f'$, the number of sampled frames $I$, the total active learning step $T$, number of stages $R$ and stage-wise budget $A_{t,r}$

**Output:** Updated video captioning model $f$

1   Initialize $f$ by training it on $\mathcal{L}$;

2   Generate frame-wise captions $\{b_j^i\}_{i=1,j=1}^{I,M+N}$ with the foundational model $f'$;

3   Parse $\{b_j^i\}_{i,j}$ into scene graphs $\mathbf{G} = \{G_j^i\}_{i,j}$;

4   **for** $t = 1$ to $T$ **do**

5      Generate predicted captions $b_j^*$ with $f$ for each video on $\mathcal{U} \cup \mathcal{L}$;

6      Parse $\{b_j^*\}_j$ into a scene graph $\mathbf{G}^*$;

7      Compute $\hat{L}_n$ based on Eq. 6 $\forall V_n \in \mathcal{U}$;

8      Ranking videos in $\mathcal{U}$;

9      **for** $V_n \in \mathcal{U}$ **do**

10         **for** $r = 1$ to $R$ **do**

11            **if** $V_n$ *belongs to the r-th region* **then**

12               Acquire $A_{t,r}$ annotations $\mathbf{C}_n^t$ for $V_n$

13            **end**

14         **end**

15         **if** $|\mathbf{C}_n^t| \neq 0$ **then**

16            $S_t \leftarrow S_t \cup V_n$;

17            $\mathbf{C}_{S_t} \leftarrow \mathbf{C}_{S_t} \cup \mathbf{C}_n^t$

18         **end**

19         **if** $\sum_t |\mathbf{C}_n^t| == D$ **then**

20            $\mathcal{U} \leftarrow \mathcal{U} \setminus V_n$;

21         **end**

22      **end**

23      $\mathcal{L} \leftarrow \mathcal{L} \cup \{S_t, \mathbf{C}_{S_t}\}$;

24      Update the video captioning model $f$ with $\mathcal{L}$;

25   **end**

---

Table 3: Results of SwinBERT on MSR-VTT test set. Specifically, SwinBert is trained with $25\%$ of training data in MSR-VTT. Among them, $Ours$ and $Random$ are active learning methods where data is chosen based on various acquisition functions. The remaining four are decided by Dataset Maps.

| | BLEU_4 | METEOR | ROUGE_L | CIDEr | SPICE |
|---|---|---|---|---|---|
| Ours | 43.86 | 29.93 | 62.57 | 55.74 | 7.39 |
| Random | 40.95 | 29.02 | 60.63 | 50.20 | 7.14 |
| Easy | 40.23 | 27.93 | 59.50 | 45.76 | 6.53 |
| Moderate | 39.76 | 27.93 | 59.71 | 46.07 | 6.74 |
| Hard | 32.29 | 25.49 | 55.66 | 31.75 | 5.84 |
| Collective Outliers | 28.92 | 25.87 | 54.39 | 29.43 | 5.71 |

ambiguity. Combining both would provide us with a way to capture data learnability. Dataset Maps, to this end, provide an interpretable way to identify collective outliers w.r.t. data learnability.

To verify our assumption that Dataset Maps helps in terms of figuring out collective outliers, we propose to divide samples according to their learnability. Specifically, samples are firstly divided into two halves based on the average SPICE score. Then the two halves are equally divided into four regions (*i.e.*Easy, Moderate, Hard, and Collective Outliers) based on variance. Each region exclusively contains $25\%$ of the samples from the full training set to avoid the interference of data size on the results. We then train $f$ with data from each region and show the overall performance on

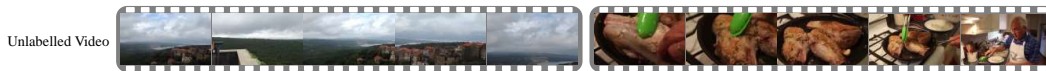

| Unlabelled Video | | |
|---|---|---|
| BLIP2 Predictions | • a drone is flying over a green forest.
• a man is flying over a lake and a city. | • a person is preparing a chicken in a pan.
• an older man is cooking in the kitchen. |
| Human-annotated GTs | • people look from an overlook toward a hillside village and a lake below on a cloudy day.
• an overlook that can see for miles lots of forest area to the left and a small town to the right. | • in a kitchen a man is frying meat in a pan turning it on the other side.
• big meat pieces after cooking are fried in the pan care to take fry both sides. |

Figure 10: Qualitative comparisons between BLIP2 predictions and human-annotated ground truths.

Table 4: Results of SwinBERT trained with different mixed MVR-VTT datasets combined with 5% ground truths (*i.e.*initial seed set) and BLIP2 captions filtered by threshold $th$.

| $th$ | 4 | 8 | 12 | 16 | 20 | 24 | 28 |
|---|---|---|---|---|---|---|---|
| Number of BLIP2 captions | 92765 | 58050 | 28880 | 12561 | 4509 | 1406 | 391 |
| CIDEr | 31.92 | 32.82 | 32.88 | 33.39 | 34.96 | 39.32 | 39.90 |
| SPICE | 7.09 | 6.99 | 6.92 | 6.65 | 6.56 | 6.24 | 5.70 |

the test set in Tab. 3. In short, We can draw the following two conclusions: (1) There exist obvious gaps between regions of various averaged SPICE scores (e.g. Easy v.s. Collective Outliers, Moderate v.s. Hard), indicating that simple samples bring in more performance boosts. (2) When the difficulty of samples is close, samples of higher variances lead to better performance (*e.g.*. Moderate v.s. Easy, Hard v.s. Collective Outliers). In our active learning scenario, this observation can be interpreted as simple and learnable samples with certain uncertainty (*e.g.*samples in the Moderate region) are of greater informativeness.

We further conduct another baseline where $f$ is trained with 25% randomly sampled data from the full training set. Compared to results from data with various learnability, we observe that this new baseline gives better performance. We argue that such inferiority comes from a lack of data diversity and uncertainty. As a reference, we also report the performance of our active learning method with 25% training data, which clearly gives the best performance with the same amount of data.

## A.4 Efforts to Combine Knowledge from LVLMs

Given the observation that LVLM predictions provide satisfactory approximations to human-annotated ground truths, one might wonder whether they can be directly used to either replace the human annotations or enrich them.

Motivated by this idea, we conduct a simple semi-supervised learning experiment where the initial setup is the same w.r.t. our active learning scheme, e.g., 5% and 95% split for labelled and unlabelled sets. For each video in the unlabelled set, we obtain its pseudo ground truth from BLIP2 by randomly selecting 16 captions from 32 frame-wise predictions. We then update our $f$ with these pseudo ground truths and report its performance on the MSR-VTT test set in Fig. 10. Apparently, there exist clear differences in language style between captions generated by BLIP2 and human-annotated ground truths. Specifically, this simple semi-supervised learning method achieves 0.31 and 0.07 for CIDEr and SPICE, which are worse than the starting points. Such performance drop might be caused by domain gaps between video and image captioning tasks, as well as the lack of ability to capture temporal information.

Can the gap between human-annotated video captions and predictions from image-based foundational models become smaller if the $V_n$ is so simple that video caption degenerates to image caption? Since there is no way to recover the temporal information from frame-wise image caption, we propose to address this factor from another perspective. Specifically, we believe that the impact of temporal information can be minimized if videos are highly coherent among frames. Such coherency can be further approximated by frame-wise consistency. To this end, we propose to include videos whose frame-wise BLIP2 captions are of high abstraction consistency. Mathematically, let's denote $P(b_n^i)$

as follows:

$$P(b_n^i) = \frac{\sum_{k \in O_n^i} H_k(\mathbf{G}_n)}{|O_n^i|} + \frac{\sum_{k \in A_n^i} H_k(\mathbf{G}_n)}{|A_n^i|}, \tag{7}$$

Then the abstraction consistency of each BLIP2 caption in unlabelled videos is defined as follows:

$$\hat{P}(b_n^i) = P(b_n^i) + \frac{\sum_{k \in R_n^i} H_k(\mathbf{G}_n)}{|R_n^i|}. \tag{8}$$

As long as its score $\hat{P}(b_n^i)$ is not lower than a threshold $th$, a BLIP2 caption will be counted as a pseudo ground truth. Later on, these pseudo ground truths will be used to re-train $f$, which is initialized with $5\%$ of training data only. We report the performance of $f$ in Tab. 4 with $th \in \{4, 8, 12, 16, 20, 24, 28\}$. Noticeably, CIDEr score increases with the increasing $th$. This observation validates our hypothesis that videos of highly consistent content generally have a smaller domain gap in terms of video and image captioning, thanks to the neglectable impact of temporal information. In contrast, including more captions generated by BLIP2 brings higher SPICE. This is mainly because of the semantic-centric property of SPICE. In short, captions that are semantically similar but grammatically different will be regarded as correct ones in SPICE.

Since semi-supervised learning is beyond the scope of our paper, we do not include more experiments in this section. Though not observing performance improvements by introducing BLIP2 predictions as pseudo ground truths, we still believe that domain adaption methods with much more delicate designs would help. We look forward to seeing more work in this direction.

## A.5 Results with ChatGPT

To validate our hypothesis that inconsistency in human annotations genuinely exists and is not merely due to subjective judgments, we utilize ChatGPT as an objective tool.

Specifically, we randomly sample 500 videos from each sub-region from MSR-VTT, and then randomly select 2 ground truth captions for each sample. We then ask ChatGPT whether it believes these 2 captions describe the same content. ChatGPT will respond with one of three answers: $yes$, $no$, or $undecided$. There are two main observations. First of all, we notice that ChatGPT can provide reasonable answers to our provided examples. Here we provide one pair of examples for each answer from ChatGPT:

- *Yes*: "two teams play soccer on a field" $\&$ "a man gives commentary for a soccer game"
- *No*: "there is a man talking in front of his computer" $\&$ "there are some tips to use your computer"
- *Undecided*: "a video showcasing a modded jeep" $\&$ "a person gets his lifted jeep stuck in the dirt"

Noticeably, based on the feedback of ChatGPT, there are $49.5\%$ and $22.5\%$ of inconsistent data and hard-to-decide among all 2000 samples respectively, indicating the severity of inconsistency in human-annotated ground truths. Moreover, when comparing the percentage of data among sub-regions, we observe a clear trend of consistency increases when moving from collective outliers to easy. For instance, $22.4\%$ and $33\%$ of captions from easy and collective outliers are believed to be consistent. This observation aligns well with our arguments and supports our hypothesis as well as motivation.

