# OpenReview forum: "Learnability Matters: Active Learning for Video Captioning"
_NeurIPS.cc/2024/Conference — NeurIPS 2024 poster_

### Official Review · Reviewer_JoyP · 2024-07-06

**Soundness:** 3
**Presentation:** 3
**Contribution:** 3
**Rating:** 5
**Confidence:** 3

**Summary:**

This work proposes an active learning algorithm via data learnability and collective outliers. The algorithm takes three complementary aspects, learnability, diversity, and uncertainty, into account. The proposed algorithm leverages off-the-shelf models (e.g., LVLM) for approximations to ground truths.

**Strengths:**

The proposed acquisition function is comprehensive, capturing learnability, diversity, and uncertainty. The first component is captured via 1) the prediction consistency of the scene graph, including objects and attributes. 2) the absolute distance between the SPICE value between the predictions from image-based LVLM and the video captioning model. The second component is captured via tf-idf based on the predictions from BLIP2. The third component is captured by shared objects, attributes, and relationships among the prediction from BLIP2 and the video captioning model
The overall design of the acquisition function is reasonable.

A comprehensive review of data learnability and collective outliers is provided.

This paper conducts extensive experiments on MSVD and MSRVTT datasets. It compares widely used baselines, including maximum entropy, maximum likelihood, core-set selection, and clustered divergence.
The experiment results are promising.

**Weaknesses:**

The usage of LVLM is debatable. If the LVLM is sufficiently powerful to generate accurate captions on videos, training a captioning model via active learning seems unnecessary. If the LVLM cannot generate accurate captions, using the output from LVLM for different components of the acquisition function is unreliable.

The proposed framework requires extensive usage of LVLM and the scene graph, which incurs significant computational costs. On the other hand, many tools, such as Mechanical Turk, are available to collect human annotation cheaply.
I would like to know if the reduction of human annotation costs justifies the extensive computational cost.

The technical innovation for each component appears incremental, and the terms in the acquisition function are heuristic, without mathematical proof.

**Questions:**

Please refer to the Weaknesses.

**Limitations:**

The authors adequately addressed the limitations in the checklist.

---

> ### Author Rebuttal · Authors · 2024-08-07
>
> Thank you for your valuable comments. We will provide more details on the motivation behind our design and include additional supportive observations in the final version. Please review our feedback below. We are happy to address any further questions during the discussion period.
>
> > 1. The usage of LVLM is debatable.
>
> + Currently, LVLMs that support video are still in their infancy. Their zero-shot predictions on complex videos can be unreliable, e.g. performance on VATEX in Tab.10 of mPLUG[1]. Similarly, directly applying BLIP[2] gives an unsatisfactory performance on VATEX. For instance, 37.4 with CIDEr metrics according to [1]. Therefore, instead of directly exploiting predictions LVLM, we exploit LVLM to 1. generate human-like captions at the same granularity so that the abstraction inconsistency can be measured 2. approach granularity inconsistency by comparing predictions from LVLM and $f$(Line 176-187). Overall, while LVLM itself may not be entirely reliable, it can provide valuable information through inter and intra comparisons.
>
> > 2. If the reduction of human annotation costs justifies the extensive computational cost.
>
> + Our additional computational costs arise from the active learning algorithm, primarily due to applying BLIP2 to the unlabelled dataset and generating scene graphs from its predictions. This process occurs only once on the unlabelled set. On our hardware, consisting of 4 RTX 4090 graphics cards with a power capacity of 2000 kWh, it takes no more than 9 minutes to run BLIP2 and 5 minutes for scene graph generation on the MSVD training dataset. In contrast, according to [3], annotating the full dataset in 2010 required hundreds of annotators, around 2 months, and less than 5000 USD in total. Even if the highest annotation efficiency claimed by the author is always followed (10K captions per day), it still takes about 5 days to finish MSVD without considering the cost of Quality Control. Therefore, we argue that active learning is more efficient in terms of both time and cost.
>
> > 3.	The technical innovation for each component appears incremental, and the terms in the acquisition function are heuristic, without mathematical proof.
>
> + Our approach incorporates learnability, uncertainty, diversity, and a caption-wise active learning protocol. To the best of our knowledge, we are the first to explore collective outliers in video captioning tasks while also formulating them as learnability in an active learning framework. Although uncertainty and diversity are common strategies in active learning, we have adapted them specifically for our video captioning setting. For example, uncertainty measures the consistency between the scene graph generated from the most confident caption by $f$ and that from LVLM (Line 229-233).
>
> References:
>
> [1] Li, Chenliang, et al. "mPLUG: Effective and Efficient Vision-Language Learning by Cross-modal Skip-connections." Proceedings of the 2022 Conference on Empirical Methods in Natural Language Processing. 2022.
>
> [2] Li, Junnan, et al. "Blip: Bootstrapping language-image pre-training for unified vision-language understanding and generation." International conference on machine learning. PMLR, 2022.
>
> [3] Chen, David, and William B. Dolan. "Collecting highly parallel data for paraphrase evaluation." Proceedings of the 49th annual meeting of the association for computational linguistics: human language technologies. 2011.
>
> Once again, your time and effort is more than appreciated.

---

> ### Comment · Reviewer_JoyP · 2024-08-13
>
> I thank the authors for preparing the rebuttal. After reading the rebuttal, I intend to keep my original rating, which is positive.

---

> > ### Author Response · Authors · 2024-08-13
> >
> > We sincerely thank you for your time and effort as a reviewer for NeurIPS 2024. We appreciate your engagement and the valuable feedback that has helped us improve our paper. We are grateful for your recognition and support of our work. If you have any additional concerns, please feel free to let us know.
> >
> > Best Regards,
> >
> > The Authors

---

### Official Review · Reviewer_fZYA · 2024-07-13

**Soundness:** 3
**Presentation:** 3
**Contribution:** 3
**Rating:** 7
**Confidence:** 3

**Summary:**

The paper presents a groundbreaking exploration of collective outliers in video captioning tasks, introducing innovative active learning algorithms and an effective caption-wise learning protocol that integrates human knowledge to enhance model performance. Despite its strengths in pioneering research and sophisticated methodologies, the paper falls short in conducting cross-dataset experiments beyond MSVD and MSR-VTT, limiting its real-world applicability. Additionally, a lack of thorough limitation analysis and minor writing issues, such as visibility concerns in figures, suggest areas for improvement in future research efforts.

**Strengths:**

- Pioneering Exploration of Collective Outliers

The paper stands out for being the first to delve into the realm of collective outliers within video captioning tasks, offering a unique perspective on this aspect that has not been extensively explored before.

- Innovative Active Learning Algorithm

The paper introduces a novel active learning algorithm that considers learnability, diversity, and uncertainty, addressing the challenges posed by collective outliers. This algorithm is designed to effectively handle inconsistencies in abstraction and granularity, showcasing a sophisticated approach to improving model performance.

- Effective Caption-Wise Active Learning Protocol

The paper presents a new caption-wise active learning protocol that efficiently integrates human knowledge, demonstrating a strategic way to leverage external input to enhance the learning process. The paper showcases state-of-the-art performances on video captioning datasets using diverse model backbones, indicating the effectiveness of the proposed approaches in significantly improving the quality of generated captions.

**Weaknesses:**

- Lack of cross-dataset experiments. The experiments are only conducted inter MSVD and MSR-VTT. This does not solve the challenges of video captioning in real scenarios. Given a small annotated dataset (like MSR-VTT) and a large unannotated dataset from the web (like WebVid or VATEX), what samples should we annotate from a large web dataset to improve performance on the small annotated dataset?

- Lack of limitation analysis.

- Some writing issues. (1) The fonts in Figure 5 can be hardly seen. Better keep font size in all figures the same.

**Questions:**

See Weaknesses.

**Limitations:**

See Weaknesses.

---

> ### Author Rebuttal · Authors · 2024-08-07
>
> Thank you for your encouraging comments. In addition to including the cross-dataset experiments and limitations in our final version, we will double-check the text and redraw some figures to enhance clarity and aesthetics. Please see our responses to your concerns below. More figures are provided in the PDF file. We are happy to address any further questions during the discussion period.
>
> > 1. Lack of cross-dataset experiments.
>
> + Due to time constraints, we were unable to complete the experiment on large-scale datasets such as VATEX or WebVid. To simulate a cross-dataset setup, we used the small annotated dataset MSVD (1,200 videos with 40 captions per video) and treated MSR-VTT as the large unannotated dataset, which includes 6,513 videos paired with 20 captions each. The results using the official SwinBERT implementation are shown below.
>
> | Method  | Data Per. | BLEU_4  | METEOR| ROUGE_L| CIDEr  | SPICE   |
> | :------      | :--------:    | :--------:    | :--------:   | :--------:   | :--------:   | :--------:   |
> | Starting Point| 0\%          | 55.7119  | 39.6953  | 75.7301 | 109.3901 | 6.9690  |
> | Random| 20\%        | 62.1464  | 42.5841  | 78.6554 | 123.2620 | 7.6284  |
> | Ours      | 20\%        | 63.6971  | 43.6921  | 79.8555 | 127.4095 | 7.7973  |
> | Ours      | 5\%          | 63.6828  | 43.3397  | 79.7258 | 126.3246 | 7.5737  |
> | Ours      | +5\%        | 63.0961  | 43.6426  | 79.9844 | 130.9551 | 7.8935  |
> | Ours      | +5\%        | 63.5148  | 43.8025  | 79.8097 | 129.6331 | 7.9311  |
> | Ours      | +5\%        | 64.8791  | 44.2499  | 80.4313 | 129.0806 | 7.7748  |
>
> + We report the overall performance on MSVD test set. "Data Per." is the percentage of human annotations on MSR-VTT. "Random" refers to the random selection baseline described in Lines 268-269 of the main paper. We also report the performance of directly selecting 20% of MSR-VTT (Row 4) and iteratively adding 5% of MSR-VTT four times (Rows 5-8). As shown in the table, our AL algorithm significantly improves the overall performance of MSVD and is a more effective choice compared to random selection. Furthermore, directly selecting 20% of data is slightly less effective than iterative selection, demonstrating the benefits of curriculum learning. Notably, the overall performance peaks at two iterations, or 10% of human annotations on MSR-VTT, according to CIDEr and SPICE. Beyond this point, the performance saturates and slightly declines. This is expected, as 20% of MSR-VTT includes 26K captions and at least 1.3K videos, which is comparable to the original training set of MSVD. Adding more data from a different dataset can degrade performance, as the training may deviate from the original dataset.
>
>
> > 2. Lack of limitation analysis.
>
> + There are several limitations that we believe are worth further exploration. Firstly, our paper only briefly touches on the relationship between curriculum learning and learnability. Beyond provoking the design of the learnability term, we believe curriculum learning can enhance the interpretability of learnability terms and even active learning. Secondly, we found that current evaluation metrics, such as CIDEr, do not always align with human evaluations. More human analysis is needed for video captioning tasks. Thirdly, we made some preliminary attempts to combine knowledge from LVLMs in a semi-supervised learning manner (Line 389-394). Although we did not see a significant improvement, we believe further efforts are warranted. Lastly, our experiments with ChatGPT-4 can be improved with more refined designs. We will include these limitations in our final version.
>
> > 3.	Some writing issues.
>
> + We appreciate the reviewer pointing out the writing issues in our manuscript. For Figure 5, we will enlarge the font size and ensure consistent font size across all figures. Additionally, we will double-check the text and redraw some figures to enhance clarity and aesthetics.
>
> We are grateful for your constructive feedback and will revise our manuscript accordingly.

---

> > ### Author Response · Authors · 2024-08-13
> >
> > Thank you for your invaluable efforts and constructive feedback on our manuscript.
> >
> > As the discussion period nears its conclusion, we are eagerly await your thoughts on our responses. We sincerely hope that our response meets your expectations. Should there be any remaining concerns or if further clarification is needed, we are fully prepared to address them as soon as possible.
> >
> > Best regards,
> >
> > The Authors

---

> > ### Comment · Reviewer_fZYA · 2024-08-14
> > **Thanks for your response.**
> >
> > My concerns are well-addressed.

---

> > > ### Author Response · Authors · 2024-08-14
> > >
> > > We are deeply indebted to you for the time and energy you devoted as a reviewer for NeurIPS 2024.  Your insightful feedback has been instrumental in enhancing our paper, and we are truly thankful for your acknowledgment of our efforts. If you have any further concerns, please don't hesitate to reach out.
> > >
> > > Best regards,
> > >
> > > The Authors

---

### Official Review · Reviewer_ZEo9 · 2024-07-15

**Soundness:** 3
**Presentation:** 3
**Contribution:** 3
**Rating:** 5
**Confidence:** 3

**Summary:**

This paper works on active learning for video captioning, i.e., filtering the training data for a video captioning dataset. The authors observed significant inconsistency in the captioning annotation, due to different captioning abstraction or granularity. These inconsistency makes the model training hard. The authors then propose an active learning framework that find captions that the model can learn, combining with diversity and uncertainty metrics. The overall model trained with 25% of the annotated captions outperforms models trained with all captions on MSVD and MSRVTT.

**Strengths:**

- The analysis of the video captioning annotation inconsistency in Figure 1 is very interesting. This paper first explored this problem, and utilize it to design an active learning framework. This workflow makes a lot of sense to me.

- Experiments support the claims. The authors compared multiple active learning baselines, and show clear improvements with the proposed model.

- The implementation details are sufficient.

- The paper is well motivated and well structured. Figure 1 and Figure 2 make the method clear.

**Weaknesses:**

- The authors only report relative performance with respect to the fully supervised baseline in the main paper, which "hides" the absolute metrics of the proposed model. Only in the supplementary Table 2, the authors report the absolute metrics on MSRVTT, which seems to be significantly below SOTA on paper with code. E.g., CIDEr of the best performing models are > 70, and the proposed model is at ~55. While I am not saying the paper needs to get SOTA performance, but this gap towards SOTA is concerning. I.e., it might be that more powerful model has less learnability issues, and the proposed model may gain less improvements. This also makes the claim in the abstract "our algorithm outperforms SOTA methods by a large margin" an overclaim.

- From Section 3.3, the authors select X% of the total captioning, rather than X% of the overall videos. Arguably, annotating two captioning for the same video can be more efficient than annotating two videos, due to shared workload in watching the video. This makes the claim of " 25% of human annotations" questionable.

**Questions:**

Overall this paper studied a very interesting observation in video captioning (inconsistency in human annotation), and propose a valid framework to utilize this observation, with reasonable gains on selected baselines. My main concern is on the low baseline performance, resulting in overclaims on performance. My current rating is a boderline accept, conditioning on the authors response to the baseline issue and willing to rephrase the overclaims.

**Limitations:**

Not discussed.

---

> ### Author Rebuttal · Authors · 2024-08-07
>
> Thank you for your constructive comments. We are pleased to clarify our claims and will update our manuscript with additional supportive results. Below are our responses to your concerns. We are committed to addressing any further questions.
>
> > 1. Low baseline performance, resulting in overclaims on performance.
>
> + We would like to clarify that the "SOTA methods" mentioned in "our algorithm outperforms SOTA methods by a large margin" refers specifically to SOTA active learning methods. We will enhance this clarification in our final version. Regarding the baseline methods, SwinBERT and CoCap were selected as they offer a good balance between accuracy and efficiency (Line 288). In contrast, other SOTA video captioning methods, such as COSA[1], require extensive pre-training. In addition, fine-tuning is also required when adapting to downstream tasks to achieve SOTA video captioning performances. To validate this, we downloaded the pre-trained mPLUG-2[2] model from its official site and applied it to the MSR-VTT test set. Without fine-tuning on the MSR-VTT training set, it achieved a CIDEr score of 24.2. In contrast, mPLUG-2 ranks first on the MSR-VTT leaderboard with a CIDEr score of 80.3 after fine-tuning, demonstrating the significant performance improvement that comes from including data from the target task. Our active learning method remains highly valuable as it focuses on selecting the most informative cases for annotation rather than requiring annotation of all data, thereby leading to a more effective fine-tuning process.
>
> > 2. The claim of "25\% of human annotations" is questionable as "annotating two captioning for the same video can be more efficient"
>
> + When reconstructing a video captioning dataset, it is common for an annotator to provide just one caption per video. Although this approach is less efficient than having multiple annotations per person per video, it ensures data quality and diversity. For instance, VATEX[3] states that "Each video is paired with 10 English and 10 Chinese diverse captions from 20 individual human annotators." (Introduction). Further details in Section 3.1.1 highlight that "Towards large-scale and diverse human-annotated video descriptions, we build upon Amazon Mechanical Turk(AMT)2 and collect 10 English captions for every video clip in VATEX, where each caption from an individual worker.” Similarly, Section 2 of MSR-VTT[4] notes, "Therefore, we rely on Amazon Mechanical Turk (AMT) workers(1317) to annotate these clips. Each video clip is annotated by multiple workers after being watched. ... As a result, each clip is annotated with 20 sentences by different workers. " Therefore, our claim of "25\% of human annotations" is both practical and meaningful.
>
> References:
>
> [1] Chen, Sihan, et al. "COSA: Concatenated Sample Pretrained Vision-Language Foundation Model." Proceedings of the Twelfth International Conference on Learning Representations, 2024.
>
> [2] Xu, Haiyang, et al. "mPLUG-2: a modularized multi-modal foundation model across text, image and video." Proceedings of the 40th International Conference on Machine Learning. 2023.
>
> [3] Wang, Xin, et al. "Vatex: A large-scale, high-quality multilingual dataset for video-and-language research." Proceedings of the IEEE/CVF international conference on computer vision. 2019.
>
> [4] Xu, Jun, et al. "Msr-vtt: A large video description dataset for bridging video and language." Proceedings of the IEEE conference on computer vision and pattern recognition. 2016.
>
> Once again, we sincerely thank the reviewer for your time and effort in reviewing our manuscript.

---

> > ### Comment · Reviewer_ZEo9 · 2024-08-13
> > **Thank you for your rebuttal**
> >
> > Thanks the authors for the rebuttal. Please make sure to add "SOTA **active learning**" in the main claim. While the authors explained why the baseline numbers are lower than the actual SOTA, it is still unclear to me why these particular baselines (SwinBERT and CoCap), rather than a more well-pretrained baseline model, are selected for experiments. I also understand it is not practical to re-do large scale experiments during the rebuttal period.
> >
> > The authors response on annotation costs make sense.
> >
> > Overall, my concerns are partially resolved. I'll keep my current borderline rating.

---

> > > ### Author Response · Authors · 2024-08-13
> > >
> > > Thank you for your thoughtful feedback and for taking the time to review our rebuttal. Your valuable suggestions have greatly contributed to improving our manuscript. We appreciate your understanding of the limitations during the rebuttal period and welcome any further concerns or questions you may have.
> > >
> > > Best Regards,
> > >
> > > The Authors

---

### Author Rebuttal · Authors · 2024-08-07

The results of our cross-dataset experiments are provided in the PDF file.

---

### Decision · Program_Chairs · 2024-09-25

**Decision:**

Accept (poster)

**Comment:**

The paper proposes a new active learning algorithm for video captioning, which considers learnability, diversity and uncertainty to select unlabelled data. Experiments on two video captioning datasets demonstrate that the proposed method outperforms prior active learning methods in this domain.

Reviewer ZEo9 initially raised concerns about the overclaim of performance, low baseline performance and the annotation cost. Reviewer fZYA highlighted the lack of cross-data experiments and limitation analysis. Reviewer JoyP questions the use of LVLM, computational costs and the overall novelty of the method. The rebuttal addressed most of these concerns, although experiments on larger sota models could not be completed within the limited rebuttal period - a point acknowledged by Reviewer ZEo9 as a reasonable constraint.

After rebuttal, all reviewers maintained positive ratings with 2 BAs and 1 Accept. The AC agrees with the reviewers’ recommendations and suggests accepting the paper. The authors should include the necessary revisions in the final version as promised in the rebuttal.